# Conventional Antifungals for Invasive Infections Delivered by Unconventional Methods; Aerosols, Irrigants, Directed Injections and Impregnated Cement

**DOI:** 10.3390/jof8020212

**Published:** 2022-02-21

**Authors:** Richard H. Drew, John R. Perfect

**Affiliations:** 1Division of Infectious Diseases, Duke University School of Medicine, Durham, NC 27710, USA; john.perfect@duke.edu; 2College of Pharmacy & Health Sciences, Campbell University, Buies Creek, NC 27506, USA

**Keywords:** antifungals, aerosols, irrigation, orthopedic cement

## Abstract

The administration of approved antifungals via unapproved formulations or administration routes (such as aerosol, direct injection, irrigation, topical formulation and antifungal-impregnated orthopedic beads or cement) may be resorted to in an attempt to optimize drug exposure while minimizing toxicities and/or drug interactions associated with conventional (systemic) administrations. Existing data regarding such administrations are mostly restricted to uncontrolled case reports of patients with diseases refractory to conventional therapies. Attribution of efficacy and tolerability is most often problematic. This review updates prior published summaries, reflecting the most recent data and its application by available prevention and treatment guidelines for invasive fungal infections. Of the various dosage forms and antifungals, perhaps none is more widely reported than the application of amphotericin B-containing aerosols for the prevention of invasive mold infections (notably *Aspergillus* spp.).

## 1. Introduction

Despite the recent introduction of new diagnostic and treatment options, clinical outcomes for many invasive fungal infections (IFIs), especially in immunocompromised hosts, remain poor. These infections include (but are not limited to) invasive candidiasis and aspergillosis. The administration of approved antifungals via unconventional methods (i.e., unapproved formulations or routes) may be employed in an attempt to minimize toxicities and/or drug interactions associated with conventional (systemic) administrations, increase drug concentrations at the infection site or address an unmet need created by a lack of a commercially available formulation. Such methods include (but are not limited to) administration via aerosol, direct injection, irrigation, topical formulation and antifungal-impregnated orthopedic beads or cement.

Novel methods to administer antifungals (most often amphotericin B deoxycholate) have been reviewed previously and detailed descriptions of the administered preparations have been given [1]. This review revealed that published data regarding such administrations were mostly of low quality, often restricted to uncontrolled case reports or case series of small numbers of patients with disease refractory to conventional therapies. Multiple confounders and inadequate formulation descriptions made attribution of efficacy and tolerability information problematic.

It is the intent of this review to provide an updated and comprehensive overview regarding the role of novel antifungal administrations in the current treatment of a variety of IFIs. Information was retrieved utilizing PubMed (National Library of Medicine) and supplemented with professional meeting abstracts within the retrieved citations. To remain clinically relevant, our review is restricted to the use of existing FDA-approved, commercially available preparations (amphotericin B, nystatin, caspofungin, anidulafungin, micafungin, ketoconazole, fluconazole, itraconazole, voriconazole, posaconazole and isavuconazole) for unapproved indications and/or unapproved routes of administration in humans (indicated by section headers). To facilitate information location, we chose to organize our review into organ systems. Emphasis will be on recent reports (i.e., published in the last 10 years) and controlled clinical trials will be highlighted. When relevant, the acceptance of unconventional antifungal therapy in consensus IFI prevention or treatment guidelines will be identified.

## 2. Use by System

### 2.1. Head and Neck

#### 2.1.1. Oropharyngeal

**Mouthwashes and Lozenges.** While not generally considered an IFI, the treatment of azole-refractory oropharyngeal candidiasis has historically led to the use of a variety of enterally administered formulations of amphotericin B. Although use of amphotericin B deoxycholate (100–200 mg administered as a 100 mg/mL suspension) has been described for treatment of such infections, its current utility is limited by the lack of an approved commercial preparation in the USA as well as the expansion of several azole formulations for systemic administration (notably voriconazole and posaconazole) [2]. Despite this limitation, a 15% compounded oral suspension formulation of amphotericin B with available stability information has been published [3]. Use of amphotericin B lozenges (10 mg) for azole-refractory oropharyngeal candidiasis is also limited by the lack of commercial availability in the USA [2].

#### 2.1.2. Otic Preparations

**Powder for insufflation**. An otic capsule administered via an insufflator bulb containing amphotericin B deoxycholate 5 mg (usually co-formulated with antibacterials, such as chloramphenicol, sulfamethoxazole, sulfacetamide or ciprofloxacin, and steroids (hydrocortisone)) for the treatment of otitis externa and otomycosis has been described. However, data are lacking to support the routine use of adding antifungal-containing combination preparations in the treatment of either acute otitis externa [4] or malignant otitis externa [5]. In 2014, guidelines published by the American Academy of Otolaryngology—Head and Neck Surgery Foundation recommend the use of preparations that are specifically FDA-approved for acute otitis externa [4]. Generally, these preparations do not contain an antifungal agent.

**Otic drops.** A 2021 Cochrane review evaluated management of otomycosis utilizing commercially available topical azole drops [6]. Azoles included in the report included eberconazole, fluconazole, miconazole and clotrimazole in various formulations (creams and solutions). However, randomized studies comparing topical azoles to no treatment or placebo are still lacking, so their role in treatment is unknown [6]. In contrast, voriconazole-containing formulations have been used in the setting of otomycosis due to *Aspergillus* spp. When combined, a case report on the administration of three drops of voriconazole 1% ophthalmic solution three to four times a day in the ear canal [7] and an uncontrolled case series (*n* = 55) describing the hourly administration of 1% topical voriconazole drops (during the daytime) for two weeks [8] in the treatment of refractory otomycosis without tympanic membrane perforation reported clinical resolution in all treated patients.

#### 2.1.3. Nasal Preparations

**Irrigation.** Initial published reports on the use of intranasal administration of antifungal-containing solutions (usually amphotericin B deoxycholate) was for the prevention of IFIs (primarily aspergillosis) in high-risk patients (such as those undergoing solid organ or hematopoietic cell transplantations (HCT) or those with hematologic malignancies at highest risk of mold infections). Such use is not currently recommended, either in patients with cancer, HCT or in solid organ transplant patient populations [9,10,11,12,13,14]. Its use has largely been replaced in these patients by systemic therapies (such as voriconazole, posaconazole and isavuconazole) whose use has been established by robust clinical trials.

Isolated case reports have been published on the adjunctive use of amphotericin-containing nasal irrigations in the treatment of fungal infections, such as *Aspergillus* [10,15]. However, such use has been discouraged due to the lack of penetration into host tissues when administered as a topical therapy [10]. In contrast, the interest in such therapy for the treatment of rhinocerebral mucormycosis has been illustrated in numerous case reports describing its use. Despite the lack of quality supportive evidence, justification for such practices has included the need to facilitate the delivery of amphotericin B to poorly perfused tissues, reductions in renal toxicity associated with systemic polyene therapy and the potential to locally stimulate host responses and provide a chemical-like debulking to augment surgical debridement [16]. Concentrations of amphotericin B deoxycholate employed in these reports generally range between 50–200 mcg/mL. In many reported cases the diluent is described as normal saline despite its known incompatibility [3]. One recent case report in a pediatric patient described the successful adjunctive use of liposomal amphotericin B 100 mcg/mL administered in total doses of 2 mg weekly with a total of six doses for treatment [17]. The advantages of lipid-based formulations over amphotericin B deoxycholate in such a setting, however, are unclear. Furthermore, the heavy use of adjunctive irrigations, systemic therapy and surgical intervention make determination of the efficacy of amphotericin B-containing irrigation impossible in most cases without the availability of adequately controlled clinical trials. In addition, it is highly likely that the adverse effects of such local administrations have been understated [18]. In our own experience, patients receiving adjunctive local polyene therapy after surgical debridement have experienced severe pain and burning associated with its administration and have required either premedication to continue with the treatment or have refused further irrigation therapy. Therefore, it is our opinion that such use should generally be discouraged until adequate studies can better define its precise role and success.

Another area in which amphotericin B-containing nasal irrigations have been utilized is as an adjunct in the treatment of chronic rhinosinusitis [19]. However, unlike its local use for many other indications, the administration of amphotericin B nasal irrigations for this use has been evaluated in comparative (often placebo-controlled) trials. Wide variations between studies have been reported in the concentrations of amphotericin B utilized (ranging from 40–3000 mcg/mL), total daily doses (3–20 mg), duration of treatment (4 weeks–6 months) and the outcome measurements employed in these studies. These studies, along with the published 2015 clinical practice guidelines for the management of adult sinusitis [20] and a Cochrane review [21,22], fail to establish sufficient evidence to support amphotericin B irrigations for this indication.

#### 2.2. Respiratory Tract

**Endobronchial instillations.** Case reports on endobronchial instillations of antifungals have generally been restricted to the use of amphotericin B deoxycholate. More recently, use of amphotericin B lipid complex [23] or liposomal amphotericin B [24] have been reported for the treatment of endobronchial aspergillosis. However, given the availability of newer treatment options (such as echinocandins and extended-spectrum triazoles) and the lack of sufficient efficacy and safety data to support their use, endobronchial instillations of antifungal agents should not be routinely used for the treatment of aspergillosis [10].

**Percutaneous/intracavitary.** Numerous case reports have described the administration of antifungals (most frequently amphotericin B 50 mg in 20 mL of D5W) via percutaneous catheter for the management of pulmonary aspergilloma. Adverse events associated with such administration include (but are not limited to) coughing, fever, headaches and vomiting [25]. While case reports also describe alternate dosage forms (such as pastes and gelatins), such reports are sparse and the justification for use of another formulation is most often unclear. Percutaneous delivery of antifungals is not currently recommended for the treatment of *Aspergillus* infections [10].

**Aerosols.** Perhaps the most widespread instance of unconventional drug delivery of an antifungal is the administration of nebulized amphotericin B formulations (most commonly amphotericin B deoxycholate (aAmBd), amphotericin B lipid complex (aABLC) or liposomal amphotericin B (aLAmB)) for the prevention of invasive fungal infections (specifically *Aspergillus* spp.). Targeted high-risk patient populations most commonly include those with cancer chemotherapy-induced neutropenia and hematogenous cell (HCT) and solid organ transplantation (SOT) recipients. As with other unconventional antifungal administrations, the goal is to deliver high antifungal concentrations of an antimold agent at the site of initial infection while minimizing adverse effects and drug interactions associated with systemic administration. Considerations for such administration include formulation, dose, timing and duration of therapy, delivery method, costs and the need for concomitant systemic antifungals. Patient-related factors that may impact drug delivery include large airway diameter (often related to patient age), inhalation technique, use of mechanical ventilatory support and presence of abnormalities in airway structure [26].

The drug delivery system (i.e., nebulizer) must deliver intact drugs, especially those formulations utilizing lipid-based preparations of a particle size small enough to reach the lower airways (ideally between 1–5 µm). The nebulizers most often utilized in clinical trials assessing drug delivery, safety and efficacy often include pressure-driven air jet or ultrasonic nebulizers. More recently, vibrating mesh nebulizers have been employed in such settings due to their efficiency and ability to deliver particles of small sizes, even when liposomal formulations are employed and in patients requiring mechanical ventilation [26]. However, there are recent concerns that ABLC is unable to be delivered efficiently using such nebulizers [27].

In vitro studies have been published to characterize the nebulization of liposomal amphotericin B [28,29,30], with one [30] comparing the impact of the nebulizer on drug particle size and another [29] documenting that the liposome was intact following aerosolization. These studies have been complemented by pharmacokinetic studies of aAmBd [31,32,33,34,35] and aABLC [36,37,38] in patients (most of them involving lung transplant recipients). Those studies evaluating systemic amphotericin B exposure have demonstrated undetectable or minimal systemic absorption from administration of both aAmB [35,39] and aABLC [37,40].

Clinical studies have reported the use of aAmBd [41,42,43] and aLAmB [44,45,46] as antifungal prophylaxes in patients with neutropenia secondary to cancer chemotherapy. The doses of aAmBd most commonly used in these trials were 5–10 mg over 10–20 min twice daily until the resolution of neutropenia [41,42,43]. Patients receive aLAmB [44,45,46], 12.5 mg over 10–30 min, either daily for two consecutive days then weekly [44], for four consecutive days and then twice weekly in combination with fluconazole [45], or twice a week beginning with the first chemotherapy cycle [46]. In consideration of these data, the 2018 European guidelines for antifungal prophylaxis in hematology patients recommend aLAmB 10 mg twice weekly in combination with oral fluconazole for patients at high risk of invasive mold infections but discourage the use of alternate formulations [47]. In contrast, both the 2018 ASCO/IDSA [9] and 2021 prevention and treatment guidelines for hematology–oncology patients published by the National Comprehensive Cancer Center [48] recommend alternate antifungal strategies.

In patients undergoing HCT, use of aABLC was evaluated in an open-label, noncomparative trial evaluating administration of 50 mg daily for four days, then once per week for 13 weeks, for a total of 17 doses in combination with oral fluconazole through post-transplant day 100 [49]. Use of aerosolized formulations of amphotericin B are not presently recommended as a fungal prophylaxis strategy in adult HCT patients [13,14]. While data are limited to justify its use in pediatric HCT patients [50], aLAmB, 12.5 mg on two consecutive days per week, has been identified as an option in this population [51].

Nebulized formulations of amphotericin B have been utilized as an antifungal prophylaxis strategy in patients undergoing lung transplantation. Use of post-operative antifungal prophylaxis with an aerosolized formulation of amphotericin (± systemic antifungals) has been reported in up to 70% of US lung transplant centers [52] with either aABLC [40,53,54,55], aLAmB [56] or aAmBd [53,57,58,59,60,61]. Published experience with solid organ transplant recipients outside of those receiving lung transplantation (such as heart or heart–lung) is limited [57]. Of note is the present need for an added systemic antifungal agent, such as fluconazole, for invasive *Candida* prevention [11,54]. Use of aAmBd (20 mg three times a day up to 25 mg/d) or aABLC 50 mg daily × 4 days, then 50 mg once weekly (usually until discharge) postoperatively is currently recommended in SOT recipients at increased risk for *Aspergillus* infections [10,11]. The optimal duration of such prophylaxis remains unknown. Those at highest risk of pathogenic mold colonization pre- or post–lung transplant, mold infections in explanted lungs, or fungal infections of the sinus [10] and patients undergoing single lung transplantation [10,11] should also be considered for systemic administration of voriconazole, posaconazole, itraconazole or isavuconazole.

The role of aerosolized formulations of amphotericin B for the adjunctive treatment of invasive fungal infections has not yet been established. Descriptions regarding adjunctive use is limited to case reports or case series in patients receiving prior and/or concomitant systemic therapy. Its potential role in combination with systemic therapy in the management of lung transplant recipients with anastomotic endobronchial ischemia or ischemic reperfusion injury due to airway ischemia diagnosed with tracheobronchial aspergillosis is addressed in recent treatment guidelines [10]. In addition, the use of aerosolized amphotericin B formulations in the treatment of allergic bronchopulmonary aspergillosis (ABPA) is not well-established. Despite descriptions of such use in patients with cystic fibrosis [62], asthma [63] and HIV [64] and in lung transplant recipients [65], one available randomized trial of aLamB as maintenance therapy for ABPA showed no effect [66].

Administration of aerosolized amphotericin B formulations has been associated with adverse effects, most notably nausea, bad taste, cough, dizziness, chest tightness, mild bronchospasm and sputum production [67]. This is most notable with aAmBd [26,53,68] and likely due (at least in part) to sodium deoxycholate as the solubilizing agent [28]. In patients with ABPA and/or asthma, aAmBd is poorly tolerated [68,69]. Overall, aABLC tolerability has best been characterized among lung transplant recipients. In one report, completion rate with aABLC prophylactic regimens was 90.2% [54]. Bronchospasm was reported in 0.2% of treatments. In HCT patients, cough, nausea, taste disturbance or vomiting occurred in only 2.2% of a total of 458 aABLC administrations [49]. In this study, 5.2% of administrations resulted in a >20% decline in forced expiratory volume in 1 s (FEV_1_) or forced vital capacity (FVC) but did not require either bronchodilators or treatment discontinuation. Finally, 38 hematology patients receiving 41 treatments of prophylactic aLAmB experienced bronchospasm at rates comparable to placebo administration [70]. Despite this finding, it was noted that coughing was significantly more common in aL-AmB patients.

Concerns regarding the potential of lipid vehicles to cause fatty infiltration, macrophage vacuolation and/or “foamy macrophage” accumulation (based on reports in animal models but not demonstrated in humans after systemic administration [71]) were further investigated in lung transplant patients receiving aABLC [72]. While such findings were more commonly observed in those receiving aABLC relative to aAmBd (31.3% and 12.8% of patients, respectively), no differences were noted in either 6-month survival or high-grade rejection in these patients. Therefore, the significance of this finding is still undetermined.

In contrast to the use of amphotericin B formulations, data on the use of aerosolized echinocandins and azoles are limited but interest is expanding with improved technologies of formulation and delivery. A case report describes the successful use of aerosolized micafungin in two lung transplant recipients for the treatment of *Scopulariopsis*/*Microascus* tracheobronchitis [73]. Among the antimold azoles, a case report described the adjunctive use of nebulized voriconazole (40 mg once daily) in a patient with cystic fibrosis for severe *Scedosporium apiospermum* pulmonary infection [74]. However, the exact method of delivery was not described in this report. In another, use of voriconazole (40 mg inhaled twice daily for 2 days) in six subjects produced a median (95% CI) plasma voriconazole concentration of 8 (4–26) ng/mL within 12 h of the last dose [75]. Evidence of a growing interest in aerosolized antifungal therapy is evidenced by the introduction and evaluation of existing (voriconazole) and newer antifungal agents in early-phase clinical development.

#### 2.3. Gastrointestinal/Intra-Abdominal

**Peritoneal Lavage.** Although the addition of amphotericin B to peritoneal dialysate fluids (yielding final drug concentrations ranging from 1–4 mg/L) has been described as an adjunct to IV administration for the treatment of fungal peritonitis, adverse effects associated with such administration include abdominal pain and chemical peritonitis. Case reports also describe the use of intraperitoneal lavage fluid containing flucytosine [76,77] and voriconazole [78]. However, the mainstay of management of fungal peritonitis is systemic antifungal therapy and peritoneal catheter removal, making administration of intraperitoneal lavage containing antifungals unnecessary and impractical in most cases of peritionitis [10,79,80].

**Selective decontamination of the digestive tract (SDD**). Although the etiology of pneumonia is rarely directly linked to *Candida* spp., the potential relationship between *Candida* colonization of the respiratory tract and other infections has created interest in the prophylactic use of nonabsorbable antifungals as part of an SDD regimen [81,82,83]. Target populations for such prophylaxis have included asthma patients receiving inhaled steroids, hematogenous stem cell transplant recipients, neutropenic cancer victims, those with hematologic malignancy, ICU patients, liver transplant recipients, mechanically ventilated pediatric patients and those undergoing GI surgery [84,85,86]. Randomized controlled trials have employed amphotericin B in doses ranging from 200–500 mg four times daily, usually in combination with other nonabsorbable antibacterials, such as polymyxin and either tobramycin or gentamicin. In some cases, nystatin was used in place of amphotericin B [87,88]. However, despite the numerous clinical studies to evaluate this relationship and its manipulation, the impact of SDD on pneumonia remains uncertain [82]. Furthermore, attempts to examine nebulized amphotericin B as an SDD also remain inconclusive [89,90].

#### 2.4. Skin and Skin Structure

**Gels, creams, lotions.** Given our focus on invasive fungal infections, the use of unconventional drug administrations for cutaneous infection is beyond the scope of this review. However, topical applications of antifungals for the management of invasive fungal infections have been described. Although sporotrichosis is generally treated with systemic azoles (such as itraconazole) [91], descriptions of the use of adjunctive topical azole therapy for mold infections include a recent case report in a patient with cutaneous *Fusarium solani* infection utilizing voriconazole 1%-containing cream with a detailed description of the formulation and stability utilizing the injectable dosage form and a commercially available topical vehicle [92].

While numerous reports describe the use of topical amphotericin B deoxycholate for the treatment of cutaneous leishmaniasis, the overall quality of the data is considered low for this indication [93]. Topical applications of amphotericin B have also been administered by washes, impregnated dressings and percutaneous infusions for the treatment of cutaneous manifestations of mucormycosis [94,95]. However, such descriptions are generally limited to case reports and are compounded by adjunctive therapy (including surgery), making it difficult to assess the impact of such local treatments.

**Irrigations.** Use of amphotericin B deoxycholate-containing irrigations has been reported for the adjunctive management of cutaneous manifestations of invasive aspergillosis [96] as well as cutaneous manifestations of cocciodomycosis [97]. However, given the increasing options for systemic therapy (most notably for treatment of invasive aspergillosis), the role of topical irrigations in such settings is questionable.

**Percutaneous Delivery.** Percutaneous administration of antifungals (most often containing either amphotericin B or nystatin) in a variety of dosage forms (including injections of antifungal medications, infusions, pastes and gelatins) for the treatment of patients with an aspergilloma have been reported. However, such administration is generally discouraged due to the lack of adequate efficacy, safety and stability data along with the availability of alternative systemic treatment options [10].

#### 2.5. Central Nervous System (CNS)

Intrathecal administration of antifungals has generally involved treatment of refractory CNS infections requiring agents with either high degrees of systemic toxicity (i.e., amphotericin B) or the inability of the agent to penetrate into the CNS in concentrations adequate to treat the infection [98]. Therefore, intrathecal administration of fluconazole, posaconazole, voriconazole, isavuconazole and flucytosine are generally excluded for consideration for such administration [98]. Preservative-free preparations are generally required/preferred, utilizing strict standards to assure aseptic procedures have been maintained.

**Intraventricular.** Due primarily to the availability of alternative systemic treatment options, administration of amphotericin B deoxycholate via intraventricular injection for the treatment of CNS infections due to *Candida* spp. is generally restricted to situations where an indwelling device (such as a ventricular shunt or external ventriculostomy drain) cannot be removed [80]. Reported doses/concentrations in such cases vary widely, ranging between 0.01 and 1 mg in 2 mL of 5% dextrose in water administered daily [80]. Due to the complications (direct toxicity and/or infections) associated with intrathecal or intracisternal administration and/or through Omaya reservoirs of amphotericin B, most treatment guidelines for the management of IFIs involving the CNS do not recommend the routine use of such administrations [10,80,99,100,101].

The intraventricular administration of amphotericin B is most commonly described within the context of treatment for coccidioidal meningitis. The most extensive discussion of such usage (including detailed instructions for preparation and administration) was published in 2017 [102]. The authors describe amphotericin B deoxycholate 50 mg reconstituted with 10 mL of sterile water for injection. When combined with additional dilutions of dextrose 5% in water, the targeted dose (0.1–0.2 mg) is produced as a suspension formulation. The co-formulation/co-administration of a corticosteroid, such as methylprednisolone, has been described in attempts to lessen the side effects associated with such administrations, but the impacts on safety and/or efficacy have not been determined in a controlled clinical trial [102]. Of note, also, is the concern for the potential for drug–drug interactions between amphotericin B and hydrocortisone [103,104]. The optimal dose, maximum dose and dosing schedules are unknown or empirical. Individual doses of 0.1 mg three times weekly advanced to 0.1 mg per week if tolerated have been described [102]. Tapering the frequency of polyene administration has also been described [102].

Case reports for shunt infections describe the use of intraventricular administration of liposomal amphotericin B for *Aspergillus* [105], *Coccidioides* [106] and *Candida* with 1 mg/day dissolved in 3 mL of 5% dextrose, and a shunt was closed for 4 h after administration [107]. Limited published experience with this preparation and the advantages over amphotericin B deoxycholate for intrathecal administration are uncertain. Case reports also describe the intraventricular administration of caspofungin for *Pseudallescheria boydii* infection with 1 mg/day and later 2 mg/day via bilateral intraventricular catheters for 19 days [108] and for *C. auris* infection, given at a dose of 10 mg through the external ventricular drain, followed by clamping of the tube for 6 h [109].

The use of intraventricular injections of amphotericin B is frequently associated with and often limited by adverse reactions, the most commonly reported of these including headaches, fever and nausea/vomiting [80,110]. Direct signs of neurotoxicity from such injections may include ophthalmoplegia, hearing loss, ataxia, paraplegia and neurogenic bladder. Though usually transient, these signs may last for hours after administration and can be permanent [110].

**Intracisternal.** Intracisternal administration of amphotericin B deoxycholate for the treatment of coccidioidal meningitis (and less frequently for cryptococcal meningitis) has been reported. Severe adverse effects have also been reported with such cisternal administration, including a report of subarachnoid hemorrhage, brain stem decompensation and subsequent death. The usual starting dose of amphotericin B deoxycholate is 0.1 mg three times a week and increased weekly, if tolerated. Detailed descriptions of dosing, titration and administration are provided elsewhere [110].

#### 2.6. Bone and Joint

**Irrigation**. Both amphotericin B deoxycholate and liposomal amphotericin B irrigations have been described as adjuncts to the treatment of fungal mediastinitis. One description included local therapy with amphotericin B (10 mg in 10 mL normal saline) during surgical debridement for the treatment of mucormycosis [111]. Use of continuous liposomal amphotericin B (100 mg in 1000 mL of D5W (100 μg/mL)) has been described in a case report as adjunctive management for mediastinal mucormycosis [112]. For the management of mediastinitis due to *Candida* spp., irrigation of the mediastinal space with amphotericin B is discouraged due to the resulting irritation of the wound [80]. As with all antifungal irrigants, it is difficult to determine the contribution to the treatment outcome due to concomitant surgical intervention and concomitant use of systemic antifungals.

Polyhexamethylene biguanide (PHMB) is a chemical most commonly used for cleaning pools and in the treatment of amebic keratitis. Case reports describe the use of an irrigant containing PHMB 0.2% for orthopedic infections due to *Fusarium* and polymicrobial infections, including *Aspergillus fumigatus*, a *Fusarium* species, *Scedosporium prolificans* and *Trichoderma* species infections [113,114]. A case of *Scedosporium prolificans* osteomyelitis in an immunocompetent child also describes the adjunctive use of PHMB [114].

**Impregnated bone cement, spacers or beads.** The adjunctive role of an antifungal-impregnated delivery vehicle (most commonly polymethyl methacrylate), such as bone cement, spacers or beads, while clinically described, has not been well-studied and remains controversial [80]. In addition to concerns regarding the potential for local adverse effects, potential for surgical complications and costs, there is a critical need to understand the antifungal’s stability, release properties and impact on material integrity [115]. In vitro studies regarding the stability, integrity and release properties of the active antifungal drug, required to assess duration and amount of exposure, are limited.

The use of antifungal agents for incorporation in cement, spacers or beads generally requires a powder formulation. Amphotericin B deoxycholate is generally preferred due to its stability with respect to heat, its spectrum of activity and its availability in powder form. Reports of the use of amphotericin B deoxycholate range from 150–1500 mg per 40 gm cement. The optimal concentration, however, is unknown. In one evaluation, while amphotericin B deoxycholate did not weaken the cement, the limited duration of drug elution and amount released created doubt as to the value of its incorporation [116,117]. Limited data are also available for liposomal amphotericin B, and it has been suggested that liposomal amphotericin B has greater amphotericin B release than amphotericin B deoxycholate but that it is compromised by the compressive strength of the vehicle [118]. Both voriconazole [119] and fluconazole [120] have also been examined for their stability in bone cement. Like amphotericin B, voriconazole is available in powder formulation. The concentration of voriconazole ranges from 200–1000 mg per 40 gm cement. However, injectable voriconazole is formulated with cyclodextrin, which may weaken the cement when administered in a significant volume [121]. Case reports have described the use of beads or cement impregnated with fluconazole [122], voriconazole [123] or amphotericin B deoxycholate [123,124,125,126,127] as adjuncts to systemic antifungal therapy. Details regarding preparation of the impregnated medium are generally lacking in these reports. However, a recent case report involving treatment of an intra-articular infection due to *Candida auris* described the use of a molded spacer consisting of 40 g Palacos^®^ cement mixed with 100 mg of heat-stable powdered amphotericin B deoxycholate [127].

**Intra-articular injection.** While older reports describe the intra-articular administration of amphotericin B deoxycholate for the management of fungal synovitis and arthritis in doses ranging between 0.05 and 20 mg (most commonly 2–5 mg), current use of such injections is generally restricted for treatment of pathogens other than *Candida* spp. [80,128,129,130]. A case report also describes voriconazole articular injection for *Fusarium solani* arthritis after bone marrow transplantation [131].

#### 2.7. Ophthalmic Administrations

Considerations for formulations (free of excipients and preservatives) generally restrict choices to those already available in injectable formulations.

**Ophthalmic solution for topical application.** Use of amphotericin B drops or solution (0.05–0.2%) has been described for treatment of fungal keratitis due to a wide variety of fungal pathogens [132,133,134,135,136,137]. While the stability of liposomal amphotericin B in such preparations has been investigated [138], its clinical advantages over amphotericin B deoxycholate are uncertain, despite animal models suggesting a potential for reduction in local toxicity.

While case reports have also described the topical ophthalmic administration of echinocandins (most frequently caspofungin 0.5%) in combination with systemic therapy, the limited stability of such preparations along with the cost of the therapy is likely to limit its utility [139,140].

As a result of the high concentrations achieved following oral administration, the need for topical application of fluconazole is significantly limited. In contrast, voriconazole’s expanded spectrum against pathogenic molds, along with significant inter- and intra-patient pharmacokinetic variability, drug interactions and adverse effects associated with systemic administration make it a more attractive azole candidate for topical application [141,142,143,144,145,146]. Topical voriconazole 1% eye drop solution is currently recognized as a treatment option for *Aspergillus* keratitis and is generally well-tolerated [10,147]. More recently, topical 1% voriconazole ophthalmic solution monotherapy was identified as a treatment option for *Acanthamoeba* keratitis, where other treatment options are limited [148,149]. In fact, topical voriconazole was reported to be comparable to the combination of topical polyhexamethylene biguanide 0.02% and chlorhexidine 0.02% in a pilot study [149].

**Intraocular injections (intravitreal, intrastromal, intracameral).** Due to high intraocular penetration, *Candida* isolates susceptible to fluconazole causing endophthalmitis are most often treated with systemic therapy [80]. In contrast, the antifungal agents noted to have activity against *Aspergillus* spp., such as amphotericin B, are considered to have limited ocular penetration [80]. As discussed previously, both amphotericin B and voriconazole can be associated with significant toxicity when administered systemically. Therefore, intraocular administration of these agents has an expanded role in the treatment of disseminated aspergillosis resulting in endophthalmitis. Administration of intravitreal amphotericin B deoxycholate (5–10 µg per 0.1 mL of sterile water) has been recommended as an adjunct to the systemic administration of voriconazole in such mold infection settings [10,80,150]. However, experience with liposomal amphotericin B (0.01 mg/0.1 mL) is limited [151,152]. Intravitreal administration of amphotericin B has been associated with retinal toxicity, notably fibrinous iritis [153], retinal or pigment epithelial toxicity [154], loss of retinal ganglion cells, vitreal inflammation, corneal edema, neovascularization and inflammation. More recently, voriconazole 100 µg in 0.1 mL sterile water or normal saline that achieves a final concentration of approximately 25 µg/mL has become the preferred agent [80,150,152,155,156,157].

Similar to voriconazole and amphotericin B, the penetration of echinocandins (notably caspofungin) into the eye following systemic administration is reported to be limited [158]. A recent case report describes an intravitreal injection of 100 μg of caspofungin in a volume 0.1 mL that was utilized repeatedly and details regarding its dilution and administration are provided in the report [159]. However, the role of intraocular injections of echinocandins for the management of fungal endophthalmitis remains uncertain and requires further investigation.

**Retrobulbar injections**. A case report has described the retrobulbar injection of amphotericin B deoxycholate as an adjunct to systemic therapy for the treatment of orbital mucormycosis [160]. However, recent guidelines on the treatment of such infections advise against such use [161].

#### 2.8. Antifungal Lock Administration

Extensive in vitro and animal model studies that are summarized in detail elsewhere [162,163] describe the impact of biofilm age, catheter material (such as polystyrene, silicone, polyurethane), antifungal choice, volume, concentration of each dose, dwell time and duration on catheter sterilization. Such treatments must be compatible with the catheter material. Therefore, the fungal pathogen and its ability to form biofilm can also significantly impact the outcome and must be overcome [162,163].

Catheter removal, which is generally considered essential to treatment success combined with systemic antifungal therapy, is required to efficiently treat catheter-related bloodstream infections [80,162,163]. In situations where catheter removal is impractical, the instillation and retention of high concentrations of antifungal agents within the catheter with the intent to sterilize in situ is known as antifungal lock administration. However, quality data available to support such practices are lacking. Published case reports usually involve intravascular catheters, but one describes employment of lock administration to a peritoneal catheter [164]. Many of these involve *Candida* spp., due to its ability to produce biofilm, but one report was of an infection due to *Malassezia furfur* [165]. Most case reports in this area are from pediatric patients, with amphotericin B deoxycholate in concentrations ranging most commonly from 2.5–5.0 mg/mL for 6–24 h/day or 6–12 h/day for 14–21 days for up to several months. All of these reports utilize prior and concomitant systemic antifungal therapy, making it difficult to assess the contribution of the antifungal lock therapy to the antifungal strategy and outcome.

Despite in vitro studies reporting the activity of lipid formulations of amphotericin B against *Candida* biofilm [166], case reports of the use of lipid-based formulations of amphotericin B (primarily liposomal amphotericin B) as an antibiotic lock solution are limited in number [167,168,169,170]. Descriptions of catheter type, lock solution, volumes, dwell times, frequency and duration varied with the variety of report. In one, liposomal amphotericin B (4 mg in 5% dextrose) and 100 U of heparin per 1.5 mL was allowed to dwell for 8 h [167]. In another, liposomal amphotericin B (8 mg/3 mL) was used successfully [169]. More recently, a pilot study in children, primarily with infected catheters due to *Candida* spp., examined the use of liposomal amphotericin B at 2 mg/mL [170]. Liposomal amphotericin B was allowed to dwell for 8–12 h before its removal for a minimum of 14 days. Lines were cleared of infection in 9/12 subjects without apparent adverse effect.

Despite the potential application of echinocandins for use in a lock solution for the treatment of catheter-related fungal infections, clinical reports of their use are sparse. One report described the successful adjunctive use of caspofungin (10 mg/3 mL in 5% dextrose allowed to dwell for 12 h per day for 2 weeks) for the treatment of *Candida lipolytica* line-related fungaemia in a 9-year-old boy [171].

#### 2.9. Genitourinary

**Creams and suppositories.** Case reports of treatment-refractory or azole-resistant vulvovaginal candidiasis have described the use of intravaginal amphotericin B creams in concentrations ranging from 0.3–10% or 50 mg/d suppository [172,173,174,175]. Flucytosine cream (17%) alone or in combination with intravaginal amphotericin B has also been described [174,175]. One of these reports describes the preparation of the combination product [175].

**Bladder irrigation.** With respect to bladder irrigation, amphotericin B deoxycholate prepared as 50 mg per liter of sterile water and protected from light was recommended in the treatment of cystitis due to fluconazole-resistant *Candida* spp. for 5 days [80]. Despite such recommendations, the quality of the data supporting such use continues to be limited. The optimal dose (concentration), duration, method of administration (continuous vs. intermittent) remain unknown [176]. It is especially problematic in patients not otherwise requiring an indwelling urinary catheter and in patients with upper tract infections. Infrequent side-effects of bladder irrigations observed for such treatment include hematuria, cramping, bladder discomfort, dysuria and burning during irrigation [176].

**Nephrostomy tube irrigation.** An irrigation solution delivered via a nephrostomy tube utilizing amphotericin B deoxycholate at concentrations of 50–100 mg/L of sterile water for irrigation has been described for the management of ureteral fungus balls due to *Candida* spp. [80]. While similar recommendations have been described for aspergillosis of the renal pelvis, current treatment guidelines specifically state that such irrigations have no role in the treatment of aspergillosis involving the renal parenchyma [10]. Alternate antifungal agents described for nephrostomy tube irrigations include fluconazole in concentrations from 10–1000 mg/L administered once to six times daily [177,178,179,180,181], anidulafungin continuous irrigation 5 mg/L administered as 500 mL/24 h/tube [182] and caspofungin administered as 50 mg in 100 mL 0.9% sodium chloride infused over 24 h for approximately 7 weeks [183]. Most of these reports are limited to case reports or case series, and concomitant systemic antifungal therapy was commonly employed. Similar to the use of amphotericin B deoxycholate as a bladder irrigant, the optimal concentration, frequency and duration of such treatment has not been adequately studied. Prior to utilization of antifungal drug bladder irrigations, healthcare providers need to consider post-infusion follow-up strategies.

## 3. Conclusions

Consistent with our previous review of these treatment strategies, many of the published data regarding such antifungal administrations are of low quality, often restricted to uncontrolled case reports and case series of small numbers of patients for disease refractory to conventional therapy (see Table 1). Most reports utilized amphotericin B in attempts to minimize drug-related toxicity while utilizing its broad spectrum of antifungal activity. However, assessment of the contribution of such therapies to treatment outcome was often limited given concomitant surgical intervention and prior and/or concomitant systemic antifungal therapy. Descriptions of the formulations and their stability are most often incomplete. Safety and tolerability information is often omitted due, in part, to the retrospective nature of reports and likely due to the multiple confounders that would impact the attribution of such reactions directly to the antifungal agent. Even more difficult to assess is the impact of publication bias that exists in reporting interventions with positive outcomes in patients who are often refractory to conventional therapies.

Despite the lack of quality evidence, the administration of conventional antifungals delivered by unconventional methods often addresses important clinical needs. One such “unmet need” is the need for antifungal drugs for rare fungal infections [184]. This is most notable in the pediatric population. The lack of commercial formulations of many antifungals outside those intended for oral and parenteral administration is also a motivating factor for such unconventional usage. Unfortunately, in most instances, the existing data and published experience do not provide robust knowledge to further our therapeutic understandings. Rather, they are most often created in desperation and inadequate attention is given to formulation, efficacy, tolerability and cost.

Existing and newer antifungals are experiencing a resurgence in interest and are featuring in investigations into the prevention and treatment of invasive infections, notably invasive aspergillosis. With such renewed interest comes the attention and investment needed to adequately determine their precise role.

## Figures and Tables

**Table 1 jof-08-00212-t001:** Summary of unconventional* application of antifungals for invasive fungal infections.

System	Antifungal	Dosage Form(s)	Indication(s)/Pathogen	Comments	Select References
Head and Neck	AmBd	Mouthwash Lozenges	Candidiasis (refractory)	Use limited by available azoles for systemic administration	[2]
	AmBd	Otic powder Otic drops	Otitis externa Otomycosis	Use limited for otitis externa	[4,5]
	VOR	Otic drops	*Aspergillus* otomycosis		[7,8]
	AmBd	Nasal irrigation	IFI prevention in high-risk patients	Use has largely been replaced by systemic therapies	[9,10,11,12,13,14]
	AmBd	Nasal irrigation	Rhinocerebral mucormycosis	Role as adjunct to systemic therapy uncertain	[16,18]
	AmBd	Nasal irrigation	Chronic rhinosinusitis	Lacks evidence to support this indication	[19,20,21,22]
Respiratory Tract	AmBd LAmB ABLC	Endobronchial instillation	Endobronchial aspergillosis	Should not be routinely used	[23,24]
	AmBd	Percutaneous or intracavitary	Pulmonary aspergilloma	Not currently recommended	[10,25]
	AmBd ABLC LAmB	Aerosols	IFI prevention in high-risk patients	Guideline acceptance varies by source and patient population	[26,31,32,33,34,35,36,37,38,40,41,42,43,44,45,46,53,54,55,56,57,58,59,60,61]
Gastrointestinal	AmBd	Peritoneal lavage	Fungal peritonitis	Catheter removal is preferred	[10,79,80]
	AmBd	Oral solution	Selective decontamination of the digestive tract	Uncertain role of antifungals in this setting	[81,82,83,84,85,86,89,90]
Skin	AmBd	Washes, impregnated dressings, percutaneous infusions, irrigations	Cutaneous leishmaniasis Cutaneous manifestations of mucormycosis	Adjunctive role to systemic therapy uncertain	[92,93,94,95]
Central Nervous System	AmBd	Intrathecal infusion (intraventricular, intrathecal, intracisternal	*Candida* spp. CNS infections Coccidioidal meningitis	Poorly tolerated Requires preservative-free preparations and strict standards to assure aseptic procedures	[10,80,98,99,100,101,102]
Bone and Joint	AmB LAmB	Irrigation	Mediastinitis due to mucormycosis or *Candida* spp.	Use in of mediastinitis due to *Candida* spp. Discouraged due to wound irritation	[80,111,112]
	PHMB	Irrigation	Rare mold infections		[113,114]
	AmBd	Impregnated bone cement, spacers or beads		Concerns regarding local adverse effects, costs, stability, release properties, and impact on material integrity	[80,115,116,117,123,124,125,126,127]
	AmBd	Intraarticular injection	Fungal synovitis	For pathogens other than *Candida* spp.	[80,128,129,130]
Eye	AmBd VOR	Ophthalmic drops	*Aspergillus* keratitis		[132,133,134,135,136,137,141,142,143,144,145,146,148,149]
	AmBd VOR	Intraocular injections (intravitreal, intrastromal, intracameral)	Fungal endophthalmitis	Requires formulations free of excipients and preservatives	[10,80,150,151,152,155,156,157]
Vascular	AmBd LAmB	Antifungal lock administration	Catheter-related bloodstream infections	Catheter removal + systemic therapy essential to cure	[80,162,163,167,168,169,170]
Genitourinary	AmBd	Bladder irrigation	*Candida* cystitis	Restricted to fluconazole-resistant *Candida* spp.	[80,176]
	AmBd	Nephrostomy tube irrigation	Ureteral fungus balls due to *Candida* spp.		[177,178,179,180,181]

* Approved antifungal by unapproved formulation and/or route of administration. ABLC, amphotericin B lipid complex; AmBd, amphotericin B deoxycholate; CNS, central nervous system; IFI, invasive fungal infection; LAmB, liposomal amphotericin B; PHMB, polyhexamethylene biguanide; VOR, voriconazole.

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
