# Peer review of "Conventional Antifungals for Invasive Infections Delivered by Unconventional Methods; Aerosols, Irrigants, Directed Injections and Impregnated Cement"

_jof, 2022, doi:10.3390/jof8020212_

Round 1
Reviewer 1 Report
The article entitled “Conventional antifungals for invasive infections delivered by unconventional methods; aerosols, irrigants, directed injections and impregnated cement “ is well researched and well written, but I do have some minor issues that can be considered for further improvement:
Comments
1.Try to minimize the use of brackets in the abstract section of the manuscript.
2.Rewrite the sentences 19-23, said sentence is bit confusing
3.Why use the terminology “dose forms’’ in place of dosage forms
4.line 30-33, cite the reference for this important statement
5.Try to avoid the similar statements in abstract and introduction section” Multiple confounders and inadequate formulation descriptions made attribution of efficacy and tolerability information problematic”.
6.line no.27,Could you explain further invasive fungal infections such as typical causative organisms, complications, reasons for treatment difficulty in the introduction section
7.line no.67 What is the type of compounded formulation (15%)
8.line no.60,Is there any information regarding the efficacy of mouthwashes and lozenges in azole-refractory oropharyngeal candidiasis
9.Please provide a comment below unconventional dosage form used for oropharyngeal fungal infection
10.line no.111-known incompatibility-please provide reference
11.Comment on current Infectious Disease Society of America (IDSA) practice guidelines on catheter-related bloodstream infections caused by Candida species
12.Efficacy of EDTA plus amphotericin B lipid complex against the Candida biofilms
13.Add few points on repurposed agents and adjunctive agents in Candida biofilms
14.line 159-Is clotrimazole a good treatment option in vulvovaginal infections?
15.Utilization of clotrimazole dusting powder in such conditions?
16.line 524-What is the temperature to be used for this reconstituted preparation
17.Is there any data regarding stability and storage conditions of these preparations?
18.line 570.what is the basis of such formulation; is such preparation are compounded taking an input from formulation scientist or not
19.This article will definitely benefit healthcare professionals; however I feel that the role of amphotericin is sometimes pressed throughout the manuscript under various sections
Author Response
REVIEWER 1
The article entitled “Conventional antifungals for invasive infections delivered by unconventional methods; aerosols, irrigants, directed injections and impregnated cement “ is well researched and well written, but I do have some minor issues that can be considered for further improvement:
Comments
1.Try to minimize the use of brackets in the abstract section of the manuscript. WE REVIEWED THE ABSTRACT, EDITED AS REQUESTED, AND BELIEVE THE REMAINING BRACKETED CONTENT IS NEEDED TO PROVIDE CLARITY/DETAIL TO THE SENTENCES.
2.Rewrite the sentences 19-23, said sentence is bit confusing DONE
3.Why use the terminology “dose forms’’ in place of dosage forms DOSAGE FORMS NOW USED THROUGHOUT
4.line 30-33, cite the reference for this important statement WE BELIEVE THIS IS OUR GLOBAL ASSERTION SUPPORTED BY THE MANUSCRIPT NOT OTHERWISE ATTRIBUTABLE TO EXTERNAL REFERENCES
5.Try to avoid the similar statements in abstract and introduction section” Multiple confounders and inadequate formulation descriptions made attribution of efficacy and tolerability information problematic”. THE STATEMENT IN THE ABSTRACT WAS ABBREVIATED
6.line no.27,Could you explain further invasive fungal infections such as typical causative organisms, complications, reasons for treatment difficulty in the introduction section WE ADDED EXAMPLES OF PROBLEMATIC IFIS. WE BELIEVE THE COMPLICATIONS AND REASONS FOR DIFFICULTY ARE ADDRESSED BY STATEMENTS REGARDING ADVERSE EFFECTS AND DRUG INTERACTIONS
7.line no.67 What is the type of compounded formulation (15%) DESCRIPTION ADDED
8.line no.60,Is there any information regarding the efficacy of mouthwashes and lozenges in azole-refractory oropharyngeal candidiasis WE BELIEVE THE AMPHOTERICIN B FORMULATIONS PROVIDE HISTORICAL PERSPECTIVE (AND ARE THEREFORE INCLUDED) BUT HAVE BEEN LARGELY REPLACED BY NEWER (CONVENTIONAL) AZOLES (POSACONAZOLE OR VORICONAZOLE) FOR FLUCONAZOLE-REFRACTORY INFECTIONS. THIS IS ALSO INCLUDED.
9.Please provide a comment below unconventional dosage form used for oropharyngeal fungal infection PLEASE CLARIFY THE RECOMMENDATION REGARDING LINE OF TEXT.
10.line no.111-known incompatibility-please provide reference DONE
11.Comment on current Infectious Disease Society of America (IDSA) practice guidelines on catheter-related bloodstream infections caused by Candida species PLEASE CLARIFY LOCATION AND NEED FOR SUCH COMMENT
12.Efficacy of EDTA plus amphotericin B lipid complex against the Candida biofilms THIS DOES NOT MEET THE CRITERIA SET FOR OUR REVIEW. ALSO UNCERTAIN HOW THIS WOULD BE INTEGRATED INTO THE PRESENT CONTENT.
13.Add few points on repurposed agents and adjunctive agents in Candida biofilms PLEASE PROVIDE EXAMPLES OF AGENTS THAT FIT WITHIN OUR CRITERIA OF COMMERCIALLY-AVAILABLE FORMULATIONS WITH AT LEAST CASE-REPORT LEVEL CLINICAL DATA
14.line 159-Is clotrimazole a good treatment option in vulvovaginal infections? THIS IS A COMMERCIALLY-AVAILABLE TREATMENT OPTION AND THUS WAS EXCLUDED FROM THE REVIEW
15.Utilization of clotrimazole dusting powder in such conditions? CLOTRIMAZOLE IS COMMERCIALLY AVAILABLE IN CREAM, INTRAVAGINAL TABLET AND POWDER (ALONG WITH OTHER TOPICAL AZOLES). THEREFORE IT DID NOT CRITERIA FOR INCLUSION IN THE REVIEW
16.line 524-What is the temperature to be used for this reconstituted preparation PLEASE CLARIFY. ARE YOU ASKING HOW THE RECONSTITUTED PRODUCT IS STORED PRIOR TO ADMINISTRATION? IF YES, SUCH DETAILS ARE NOT PROVIDED BY THE REPORT.
17.Is there any data regarding stability and storage conditions of these preparations? IN GENERAL, NO. WE CITE SUCH LACK OF DATA AS A SIGNIFICANT LIMITATION TO ADOPTING SUCH PRACTICES.
18.line 570.what is the basis of such formulation; is such preparation are compounded taking an input from formulation scientist or not WE CITE SUCH LACK OF DATA AS A SIGNIFICANT LIMITATION TO ADOPTING SUCH PRACTICES
19.This article will definitely benefit healthcare professionals; however I feel that the role of amphotericin is sometimes pressed throughout the manuscript under various sections AMPHOTERICIN IS OFTEN THE “TARGET” OF SUCH PRACTICES DUE TO SPECTRUM, HISTORIC AVAILABILITY, AND TOXICITIES. THEREFORE, IT APPEARS THROUGHOUT.
Reviewer 2 Report
The review article by Drew and Perfect descrbes in detail the different unconvential ways that approved antifungals are administered in treating Invasive Fungal Infections (IFI). The article is very well written.
1)The reviewer is curious whether the authors described the administrative ways in a definite order. In other words are the different ways of drug administration described starting with the most common ones then following to the least?
2)Line 57: What does the author mean by "Used By System"?
3)Are there any reports describing the adverse effects of these kinds of antifungal administrations?
4)Are there any reports stating how sucessful are each of these techniques are in treating IFI? The authors may consider putting the success% in the table if there are any described.
Author Response
REVIEWER 2
The review article by Drew and Perfect descrbes in detail the different unconvential ways that approved antifungals are administered in treating Invasive Fungal Infections (IFI). The article is very well written.
1)The reviewer is curious whether the authors described the administrative ways in a definite order. In other words are the different ways of drug administration described starting with the most common ones then following to the least? WE CHOSE TO ORGANIZE THE DISCUSSION BY BODY SYSTEM SO THE READER COULD NAVIGATE THE LENGTHY TEXT WITHOUT AN INDEX. WE CONSIDERED OTHER WAYS TO ORGANIZE, BUT FELT THE ALTERNATE METHODS WOULD REQUIRE MATERIAL TO REPEAT/OVERLAP BETWEEN SECTIONS TO PLACE SUCH USE INTO ITS CONTEXT.
2)Line 57: What does the author mean by "Used By System"? SEE COMMENT ABOVE
3)Are there any reports describing the adverse effects of these kinds of antifungal administrations? WE GENERALLY REPORT ADVERSE EVENTS WHEN REPORTED BY THE INVESTIGATOR
4)Are there any reports stating how sucessful are each of these techniques are in treating IFI? The authors may consider putting the success% in the table if there are any described. MOST OF THESE ARE CASE REPORTS WHICH DO NOT PERMIT SUCH SUMMARY DATA. DUE TO THE UNCONTROLLED NATURE AND CONFOUNDERS PRESENT IN MOST OF THE REMAINING TREATMENT DESCRIPTIONS, WE FEEL THAT IT WOULD BE MISLEADING TO SUMMARIZE SUCH INFORMATION WITHIN THE TABLE.